# Characterization of the Common *Japonica*-Originated Genomic Regions in the High-Yielding Varieties Developed from Inter-Subspecific Crosses in Temperate Rice (*Oryza sativa* L.)

**DOI:** 10.3390/genes11050562

**Published:** 2020-05-18

**Authors:** Jeonghwan Seo, So-Myeong Lee, Jae-Hyuk Han, Na-Hyun Shin, Yoon Kyung Lee, Backki Kim, Joong Hyoun Chin, Hee-Jong Koh

**Affiliations:** 1Department of Plant Science and Research Institute for Agriculture and Life Sciences, and Plant Genomics and Breeding Institute, Seoul National University, Seoul 08826, Korea; rightseo@hotmail.com (J.S.); olivetti90@korea.kr (S.-M.L.); nknglee2403@snu.ac.kr (Y.K.L.); uptfamily@hanmail.net (B.K.); 2Department of Southern Area Crop Science, National Institute of Crop Science, RDA, Miryang 50424, Korea; 3Department of Integrative Biological Sciences and Industry, Sejong University, 209, Neungdong-ro, Gwangjin-gu, Seoul 05006, Korea; 0724jh@gmail.com (J.-H.H.); skgus1125@gmail.com (N.-H.S.)

**Keywords:** rice, yield, HYV, Tongil, *indica*, *japonica*, SNP, molecular breeding

## Abstract

The inter-subspecific crossing between *indica* and *japonica* subspecies in rice have been utilized to improve the yield potential of temperate rice. In this study, a comparative study of the genomic regions in the eight high-yielding varieties (HYVs) was conducted with those of the four non-HYVs. The Next-Generation Sequencing (NGS) mapping on the Nipponbare reference genome identified a total of 14 common genomic regions of *japonica*-originated alleles. Interestingly, the HYVs shared *japonica*-originated genomic regions on nine chromosomes, although they were developed through different breeding programs. A panel of 94 varieties was classified into four varietal groups with 38 single nucleotide polymorphism (SNP) markers from 38 genes residing in the *japonica*-originated genomic regions and 16 additional trait-specific SNPs. As expected, the *japonica*-originated genomic regions were only present in the *japonica* (JAP) and HYV groups, except for Chr4-1 and Chr4-2. The *Wx* gene, located within Chr6-1, was present in the HYV and JAP variety groups, while the yield-related genes were conserved as *indica* alleles in HYVs. The *japonica*-originated genomic regions and alleles shared by HYVs can be employed in molecular breeding programs to further develop the HYVs in temperate rice.

## 1. Introduction

There are two subspecies in cultivated rice (*Oryza sativa*), *indica* and *japonica*. *Indica* rice is known to be adaptable to tropical regions, while *japonica* rice is grown in temperate regions. Therefore, *indica* and *japonica* have different characteristics [1]. In general, *japonica* varieties are known to have relatively low yield potentials, as compared to *indica* varieties. To improve the yield potential of *japonica* rice, inter-subspecific crosses between *indica* and *japonica* have been conducted by conventional rice breeders [2]. As a result of these efforts, several high-yielding varieties (HYVs) have been developed from *indica*-*japonica* crosses. One of the greatest historical successes of *indica*–*japonica* crosses was the development of Tongil, an HYV, in Korea. Tongil showed a 30% higher yield than those of the conventional *japonica* varieties. By growing Tongil rice, self-sufficiency in staple food in Korea was possible in 1977 [3]. There were important trade-offs in growing Tongil, such as cold intolerance, pathogen susceptibility, and low eating quality, which were inherited from *indica* parents. A series of ‘Tongil-type’ HYVs have been developed from *indica*–*japonica* crosses to overcome the vulnerable points of Tongil from the late 1970s [2].

HYVs have been developed in Japan using *indica* and *japonica* varieties since the 1980s. Takanari is a Japanese semi-dwarf HYV, developed from the crosses between Milyang 42 and Milyang 25. Takanari shared the ancestry of Tongil [4]. To date, it recorded the highest grain yields for both yield trials (>10 t/ha as brown rice) and individual trials (11.7 t/ha as brown rice) in Japan [5]. Minghui 63, which was derived from the cross between IR 30 and Gui 630, is the male parent of the elite hybrid rice Shanyu 63 from China. IR 30 is a semi-dwarf variety developed in IRRI (International Rice Research Institute) and is a restorer line for WA-CMS A-lines, which have a good plant type, a high resistance to blast, a bacterial blight, and brown planthoppers. Gui 630 is a rice germplasm from Guyana that has a high grain weight, desirable grain quality, and high yield potential [6]. Gui 630 is known as an *indica* restorer variety [7]. Minghui 63 was classified into the *indica II* subpopulation, together with Milyang 23 and some other *indica-japonica* HYVs, by genome sequence analysis [8].

Nipponbare is the *japonica* reference genome of rice and was first sequenced at the high-quality whole-genome level through all the crops [9]. In addition, the whole genome sequences of *indica* rice varieties were reported [10,11,12,13,14]. The genomic difference between *indica* and *japonica* at the sequence level has been extensively studied [15]. At least 384,431 single nucleotide polymorphisms (SNPs) and 24,557 insertion/deletion mutations (InDels) were reported between Nipponbare and 93-11 [16]. With the advent of Next-Generation Sequencing (NGS) technology, numerous genomes of diverse rice germplasm collections have become available. For instance, 3000 rice genomes were sequenced and deployed in genetic and genomic rice studies [17,18,19,20]. Recently, more than one hundred high-yielding loci, associated with green revolution phenotypes and derived from the two ancestral *indica* varieties, were identified with the help of pedigree analysis, whole-genome sequencing, and genome editing [21]. Furthermore, most of the quantitative trait loci (QTLs) and genes for high-yielding potential in HYVs originated from *indica* parents in previous studies using HYVs derived from *indica*–*japonica* crosses [3,4,5,22,23]. No report exists on the characterization of *japonica* genomic regions in HYVs derived from *indica*–*japonica* crosses yet.

Previously, we sequenced the whole genomes of Tongil and its parental varieties to analyze the genome composition and genetic factors of Tongil. As a result, the Tongil genome was found to be derived mostly from the *indica* genome, with a small portion of *japonica* genome introgressions [3]. This study was carried out to comparatively analyze the genome structure of eight HYVs and to identify the *japonica*-originated genomic regions that are shared in HYVs, which will be helpful in understanding the role of *japonica* genome in Tongil and other HYVs that are developed in temperate regions for the further development of promising HYVs.

## 2. Materials and Methods

### 2.1. Plant DNA Materials

The eight HYVs, including Cheongcheongbyeo, Dasanbyeo, Hanareumbyeo, Milyang 23, Minghui 63, Nampungbyeo, Takanari, and Tongil, and four non-HYVs, including Nipponbare, Yukara, IR 8, and TN1 were used for the whole-genome sequence analysis. Cheongcheongbyeo, Dasanbyeo, Hanareumbyeo, Milyang 23, and Nampungbyeo are Tongil-type HYVs that were developed in Korea. Takanari and Minghui 63 are HYVs from Japan and China, respectively. The pedigree of each of the eight HYVs can be found in Appendix A. A total of 94 rice varieties were used for SNP marker validation (Appendix A).

### 2.2. Whole Genome Sequencing and DNA Variation

Tongil and its three parental varieties (Yukara, IR 8, and TN1) were sequenced in a previous study [3]. The other eight varieties were sequenced in this study using the Illumina Hiseq 1000 and NextSeq 500 platform (Illumina, San Diego, CA, USA) (Table 1). Whole genome sequencing, including the construction of shotgun DNA libraries, was performed according to the methods recommended by the manufacturer. The Illumina whole-genome shotgun paired-end DNA sequencing data were filtered to obtain high-quality sequence data. Raw sequence reads were subjected to quality trimming using FastQC v0.11.3 (http://www.bioinformatics.babraham.ac.uk/ projects/fastqc/), and the reads with a Phred quality (Q) score <20 were discarded. Adapter trimming was conducted by using Trimmomatic version 0.36 (http://www.usadellab.org/cms/?page=trimmomatic).

The clean reads were mapped on the *japonica* reference Nipponbare genome (Os-Nipponbare-Reference-IRGSP-1.0 [24]) using the Burrows–Wheeler Aligner (BWA) program version 0.7.15 [25]. The alignment results were merged and converted into binary alignment map (BAM) files [26]. The BAM files were used to calculate the sequencing depth and to identify SNPs using the GATK program version 3.4 with default parameters [27].

All the raw sequence data obtained in this study are available in the NCBI Short Read Archive (SRA) database under the following BioProject accession numbers: Nipponbare [PRJNA264254], Milyang 23 [PRJNA264250], Dasan (or Dasanbyeo) [PRJNA222717], Cheongcheong (or Cheongcheongbyeo) [PRJNA616202], Nampung (or Nampungbyeo) [PRJNA616219], Hanareum (or Hanareumbyeo) [PRJNA616209], Takanari [PRJNA616222], and Minghui 63 [PRJNA616216]. The raw sequence data of Tongil, Yukara, IR 8, and TN1 are available in a previous study [3].

### 2.3. SNP Allele Calling

Genotype calling was performed in order to identify SNPs that originated from the *indica* and *japonica* genomes. There were two types of values calculated in this study: variety value and reference value. The variety value calculated whether it was of *japonica*-type parental allele (Yukara allele) of Tongil or a Tongil-like variety or not. The variety value of SNP was calculated as the sum of the following values: ‘1’ (IR 8 allele); ‘2’ (TN1 allele); ‘4’ (Yukara allele); and ‘0’ (all others). If the SNP is the same as the Nipponbare reference, the value ‘1’ was given to the SNP; otherwise, the value ‘0’ was given. The SNPs showing variety value ‘4’ and reference value ‘1’ were called *japonica*-type SNPs. Then, the total number of *japonica*-type SNPs in each 100 kb block, which is the approximate chromosomal distance for the linkage disequilibrium (LD) decay rate in rice [28], in each chromosome was counted to identify the introgressed regions of *japonica*.

### 2.4. SNP Marker Development for the Fluidigm Platform

A total of 39 representative genes were selected from the selected regions. The representative genes were well-annotated in the public gene/QTL databases, as of 6 January 2017 (RAP and UniProt). Then, only the genes containing non-synonymous SNPs in the predicted exon and UTR regions between HYVs and non-HYVs were selected. We assumed that these genes might be a part of the candidate genes that are associated with *japonica*-originated traits. Out of the many SNPs in the genes, only one SNP with a substitution polymorphism between *indica* (IR 8 and TN1) and *japonica* (Nipponbare and Yukara) per representative gene was selected for the genomic validation. The SNP markers for the Fluidigm platform (Fluidigm Corporation, San Francisco, CA, USA) were designed using the method by Seo et al. [29]. To design Fluidigm SNP genotyping assays, 60–150 bp sequences, flanking the selected SNPs on either side, were aligned by BLAST. Finally, the selected SNPs and flanking sequences were uploaded on the D3 Assay Design website [30]. After confirming the results, the designed assays were ordered. One Fluidigm SNP assay contained the Allele-Specific Primer 1 (ASP1), ASP2, Locus-Specific Primer (LSP), and Specific Target Amplification (STA) primer.

### 2.5. DNA Extraction and Fluidigm Genotyping

Young leaves of each plant from all materials used in this study were collected for DNA extraction at the tillering stage. Genomic DNA was extracted using the modified cetyltrimethylammonium bromide (CTAB) method, as described by Murray and Thompson [31]. The concentration and purity of DNA samples were measured with a NanoDrop 1000 spectrophotometer (NanoDrop Technologies, Wilmington, DE, USA). DNA samples, showing absorbance ratios above 1.8 at 260/280 nm, were diluted to 50 ng/μL and used for genotyping.

Genotyping was performed using the BioMark™ HD system (Fluidigm) and 96.96 Dynamic Array IFCs (Fluidigm), according to the manufacturer’s protocol in National Instrumentation Center for Environmental Management (NICEM), Seoul National University (Pyeongchang, Korea). Specific Target Amplification (STA) was performed prior to SNP genotyping analysis. PCR was performed in a 5 μL reaction containing 50 ng of the DNA sample, according to the manufacturer’s protocol. For genotyping, SNPtype assays were performed using STA products, according to the manufacturer’s protocol. The genotyping result was acquired using Fluidigm SNP Genotyping Analysis software. All the genotype-calling results were manually checked and any obvious errors in the homozygous or heterozygous clusters were curated.

### 2.6. Data Analysis and QTL Comparison

We analyzed basic marker statistics, such as major allele frequency (MAF), heterozygosity, and polymorphism information content (PIC) of SNP markers using PowerMarker V3.25 [32]. PowerMarker V3.25 was used to calculate the genetic distance, based on CS Chord [33] and the constructed un-weighted pair group methods with the arithmetic mean algorithm (UPGMA) dendrogram, which was visualized in Molecular Evolutionary Genetics Analysis version 7.0; MEGA7 [34].

We extracted QTL information from the Q-TARO database at 25th August 2017. The physical position in Q-TARO is based on IRGSP build 4 of the Nipponbare genome, while IRGSP 1.0 of the Nipponbare genome was used in this study. Thus, the physical position of the start and end of each QTLs in Q-TARO were converted into the physical positions of IRGSP 1.0. Then, the QTLs overlapped on common *japonica*-originated regions of eight HYVs were selected. After checking if all the selected QTLs were unique or redundant, the redundant QTLs were discarded and only the unique QTLs were left remained. After the filtering step for the QTLs, they were classified into seven categories. The category classification was conducted by checking the characters and the trait names manually.

## 3. Results

### 3.1. Whole Genome Sequencing and SNP Calling

To analyze the genomic composition of the HYVs derived from *indica*–*japonica* crosses, the whole genomes of HYVs and four varieties were sequenced on the Illumina platform. A large number of short reads were mapped onto the reference Nipponbare genome and were then assembled into a consensus sequence. The number of sequence yields and the number of reads and mapping depths were varied. For example, a total of 66,464,246 reads of the Cheongcheongbyeo genome, corresponding to 9,991,040,272 bp (10 Gb), were generated, representing a 21-fold mapping depth. Nipponbare showed the largest number of sequence yields and number of reads and mapping depths (Table 1).

More than one million SNPs were detected in each of the eight HYVs and two *indica* varieties against the Nipponbare (*japonica*) genome. More than 90% of the SNPs were detected in the intergenic region. The smallest number of SNPs were in the 5′ untranslated region (UTR). Two *indica* varieties, IR 8 and TN1, represented a relatively large number of SNPs, as compared to seven HYVs except for Minghui 63. Among the seven HYVs, Milyang 23 showed the smallest number of SNPs (Table 2). 

### 3.2. Evaluation of Japonica-Type SNP Value

To identify the SNPs that originated from the *indica* and *japonica* genomes, the SNP value was evaluated. The SNPs showing a variety value of 4 and a reference value of 1 were *japonica*-originated SNPs. The total number of *japonica*-originated SNPs in each 100 kb block of each chromosome were counted in order to identify introgressed regions from *japonica*. Previously, we discriminated the Tongil genome into segments that originated from *indica* and *japonica* using the sliding window method [3]. In this study, we allowed for two exception from the HYV in order to include Takanari, Cheongcheong, Nampung and Minghui 63. Accordingly, a total of 14 *japonica*-originated genomic regions, which were shared by at least six HYVs on *japonica*-originated segment of Tongil, were detected. The common *japonica*-originated genomic regions were distributed on nine chromosomes, not including chromosomes 8, 10, and 12. There were three *japonica*-type regions on chromosome 2 and two regions on chromosomes 1, 4, and 7. Furthermore, the regions were clustered or closely located on each chromosome. The size of the regions was varied from 0.1 Mb for Chr7-1 and Chr11-1 to 2 Mb for Chr1-2. Out of the 14 regions, seven were common in eight HYVs (Figure 1, Table 3).

### 3.3. QTL Comparison and Representative Gene Selection in Common japonica-Originated Genomic Regions

To elucidate the function of common *japonica*-type regions in HYVs, we first investigated the reported QTLs in the Q-TARO database [35]. A total of 101 selected QTLs for seven categories were co-located, with 14 common *japonica*-type regions on nine chromosomes. Category classification was carried out by checking the character and trait names of each QTL manually. For instance, the yield-related trait category contained various characters which could affect yield potential, such as source activity- and sink-related morphological traits, and sterility. Only three regions on chromosome 2 were co-located with QTLs for all the seven trait categories. For eating quality, abiotic stress, and the yield-related category, 80 QTLs were identified. The largest number (10) of co-located QTLs was detected in the Chr6-1 region for eating quality. All the regions were co-located with the QTLs for abiotic stress tolerance (Table 4). This information of co-located QTLs, with common *japonica*-type regions, suggests that common genomic regions in HYVs might be mainly associated with quality, yield, and abiotic stress tolerance.

Furthermore, we selected 39 genes containing non-synonymous SNPs, which could affect the molecular function of genes, and are clearly annotated in the databases of 12 common *japonica* chromosomal introgressions. There was no target gene that satisfied the above-mentioned condition in Chr4-2 and Chr7-1. The largest number (13) of selected genes was located on Chr1-2, which is the largest region, spanning 2 Mb. Only one gene was selected from Chr2-2, Chr7-2, and Chr11-1. The size of these three blocks was 0.1–0.2 Mb. The genes annotated from the major criteria of interest were *Os01g0348900, Os06g0130000, Os06g0130100* (stress tolerance), *Os06g0130400* (eating quality), and *Os01g0367100* (yield potential) (Table 5).

### 3.4. SNP Marker Development and Genotyping Using Fluidigm Platform

A total of 39 SNP markers were designed in the 39 selected genes from common *japonica*-originated genomic regions, by one marker per one gene. The SNPs for the marker were selected from among the non-synonymous SNPs. Five SNP markers, out of the 39 SNP markers, were designed in 3′ or 5′ UTR. In addition, 14 agronomic traits, related to SNP markers in *indica*-*japonica* SNP set 2 [29] and four previously developed yield related SNP markers [36], were also used for the genotyping of 94 diverse germplasms. A total of 57 SNP markers were genotyped for 94 germplasms using the Fluidigm system, and consequently, 54 SNPs showed polymorphism and a clear genotype, except one monomorphic SNP marker designed in *Os01g0348900* on the Chr1-2 block and two SNP markers which showed low base call quality, SaF-CT and SLG7-GC, in *indica*-*japonica* SNP set 2. Therefore, we conducted a further analysis using a total of 54 polymorphic SNP markers (Figure 2, Appendix A).

The results of genotyping showed a dividing pattern for 94 varieties. A phylogenetic analysis of 94 varieties was carried out using 54 polymorphic SNP markers. There were four groups, including IND1, IND2, HYV, and JAP, in the phylogenetic tree (Figure 3). All the sequenced HYVs, except Takanari, which possesses *indica* allele of Chr1-1 and Chr1-2, were clustered in the HYV-group with seven Tongil-type and five *indica* varieties. These varieties were developed by inter-subspecific crosses or by inter-varietal crosses including the parents of Tongil-type varieties. For example, Keunseombyeo was derived from a cross between Dasanbyeo and Namyeongbyeo [37]. Taebaekbyeo was used for the development of Hanareumbyeo and Dasanbyeo (Appendix A). On the other hand, among the five *indica* varieties, four *indica* (IR24, IRBB23, IRBB61, and IRBB66) were developed by the International Rice Research Institute (IRRI). IRBB23, IRBB61, and IRBB66 are near-isogenic line series of IR24 for the improvement of the resistance against bacterial leaf blight [38]. IR24 is a good eating quality and high-yielding variety developed from the inter-subspecific crosses using one tropical *japonica* (CP-SLO) and two *indica* (SIGADIS and IR8) [39]. Furthermore, IR24 was included in the breeding program of six HYVs, which were used for resequencing in this study. (Appendix A). This implies that the specific genomic regions were conserved in the HYV group by the selections during the conventional breeding programs for HYV development.

A total of 38 SNP markers were developed in the common Tongil-like *japonica*-like regions and discriminated IND1, IND2, and HYV by the frequency of *japonica*-alleles. The HYV group showed 93.8% for *japonica*-alleles frequency, which is similar to that of JAP. A total of 16 SNP markers were related to yield and some agronomic traits could distinguish JAP from the other three groups. For the 16 SNP set, the JAP group represented 86.1% of *japonica* allele, while the other three groups showed a lower *japonica* allele frequency, which was less than 30%. The HYV group contained *indica* alleles and were informed by the makers that were linked to the genes associated with plant architecture (SD1-GA, NAL1, and TAC-CT), yield potential (GIF1, Hd6-AT, Ghd7, and GW8-AG), and subspecies differentiation (Rd-GA, qSH1-TG, and S5-TC). They contained more than 50% japonica alleles using the markers linked to the genes for grain shape and quality (GRF4, GS3-CA, qSW5-AG, GS6-GT, and WAXY-TG) of HYV-type. Practically, the markers designed in the *japonica*-originated genomic regions and the yield-related markers from *indica* varieties can differentiate HYV from *indica* and *japonica* varieties. In fact, a high proportion of *japonica* alleles on Chr1-1, Chr1-2, and Chr3-1 were found in IND2, which consist of three *aus* varieties, one wild rice relative accession, and one weedy rice variety.

## 4. Discussion

All eight HYVs used in this study were clustered into the *indica* group, based on *indica*–*japonica* SNP sets reported in a previous study [29]. Interestingly, some *japonica*-type SNPs were detected within the genomes of HYVs after resequencing analysis. Furthermore, collocations of 14 *japonica*-originated genomic regions commonly present in Tongil-like HYVs, inherited from *indica*–*japonica* crosses, were investigated. This suggests that these *japonica*-type genomic segments were commonly and repeatedly selected during independent breeding programs in the temperate rice cultivation area. To investigate the role of these *japonica* segments in HYVs, a comparative study of the reported QTLs and representative gene selection were conducted. Eating quality, stress tolerance, and yield related traits might be main drivers for the selection of the rice HYV breeding program.

A total of 54 SNP markers, including 38 SNP markers developed from 38 selected genes for 14 common *japonica*-type regions and 16 trait-specific SNP markers, were used for the genotyping of 94 diverse varieties across *indica* and *japonica*. For the 38 SNP marker set, 16 SNP markers were located on chromosome 1, which was not identified as a *japonica* genomic segment in Takanari. Consequently, Takanari was clustered into the IND1 subgroup, although it showed a similar genotypic pattern to the other HYVs. The Chr1-1 and Chr1-2 regions were co-located with some abiotic tolerance and yield related QTLs. Furthermore, there were several selected genes that conferred abiotic stress tolerance and yield potential in the blocks. For instance, *Os01g0337100* (*OsTPS1*) reported an association with abiotic stress response and tolerance by knock-out and overexpression [40,41]. *Os01g0367100* (*PHD1*) was shown to be a gene that was involved in galactolipid biosynthesis and affected photosynthetic efficiency [42]. Recently, the effect of the haplotype of *PHD1* on grain yield was also reported using the 3K rice genome panel [18]. Takanari could not have acquired this *japonica* genomic segment from a different natural environment and/or breeder’s selection.

In addition, 18 varieties out of the 19 HYV-types, showed *japonica*-type *Wx^b^* allele on the SNP marker WAXY-TG, which was designed on the splicing site in the intron of *Wx* gene in the *japonica*-type region. The *Wx* gene only contains synonymous SNPs, although it is located within Chr6-1; thus, it was not selected in our study. We previously developed a functional SNP marker for the *Wx* gene [29]. The genomic region containing the *Wx* gene is a hotspot for grain quality [43] and has been selected during and after the domestication of rice [44]. The other genomic research using two Tongil-type varieties also showed a *japonica*-type SNP pattern on the common *japonica*-type region on chromosome 6 [45]. Tongil-type varieties showed medium amylose contents, approximately 19–20%, which is similar to that of non-waxy Korean *japonica* varieties [46,47]. Further, *Os06g0130400*, one of the selected genes in Chr6-1, was also reported as the gene controlling starch grain size in endosperm [48]. *Os06g0130000* and *Os06g0130100* were reported for resistance to rice blast and bacterial blight, and tolerance to drought and salt stress, respectively [49,50]. Therefore, Chr6-1, including the *Wx* gene, might be mainly selected for eating quality and latent stress tolerance.

When HYVs were developed by inter-subspecific hybridization, the breeders aimed at not only transferring some of the desirable characteristics, like resistance to lodging, blast, and yield, but also at retaining the ecological adaptability and eating quality of *japonicas* [51]. The *japonica* chromosomal introgression regions identified in this study were regarded as putative temperate region adaptable and improved the eating quality of *indica*. For this reason, the varieties developed by *indica*–*japonica* crosses could also be considered as ‘temperate *indica*’. Recently, new elite rice varieties showing high yield potential and high grain quality were developed by the precise pyramiding of major genes controlling yield and grain quality traits [52]. Furthermore, there was an attempt to develop cold tolerant *indica* using an inter-subspecific cross and marker-assisted selection (MAS) [53]. In other words, breeding *indica* varieties, which are adaptable to the temperate region with high yield potential and good eating quality, can be efficiently achieved through inter-subspecific crosses and marker-assisted selection using the SNP markers developed in this study. Nevertheless, to dissect the exact contribution of *japonica*-type regions in HYVs, a comprehensive genetic and physiological analysis, by applying the molecular markers developed in *japonica*-type regions to the segregation populations derived from cross between HYVs and *indica*, is necessary. In addition, the functional studies of genes in the regions, as well as the selected ones in this study, are also required.

## 5. Conclusions

Consumer preference in grain shape and quality during conventional breeding procedures, without sacrificing the high-yield potential of *indica*, were revisited by the genomic analysis of HYVs. The 14 *japonica*-originated genomic regions and alleles identified in this study shared by HYVs could be applied in further development of more HYVs through inter-subspecific rice breeding in temperate rice.

## Figures and Tables

**Figure 1 genes-11-00562-f001:**
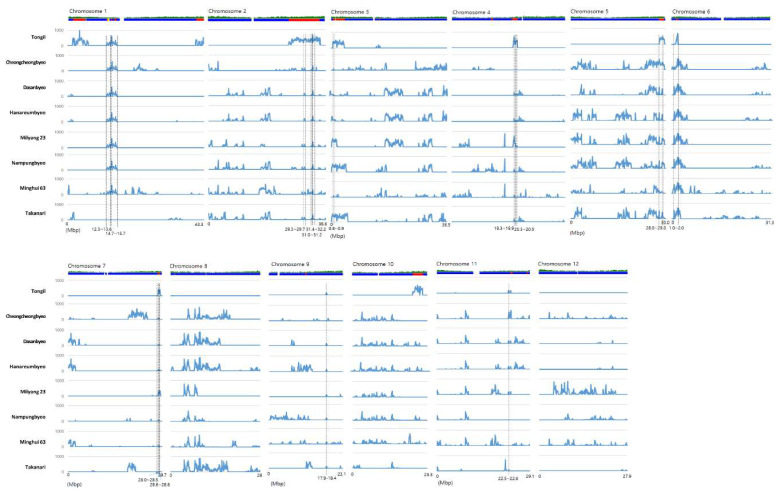
Location of *japonica*-type SNPs on 12 chromosomes of HYVs and their co-location with the *japonica* block of the Tongil genome, shown as a bar above the graphs. The blue and red block of the Tongil genome represents the *indica* and *japonica* blocks, respectively. The blue peak on each graph indicates the number of *japonica*-type SNPs. The position and range for the co-location of *japonica*-type blocks are denoted by vertical black dotted lines.

**Figure 2 genes-11-00562-f002:**
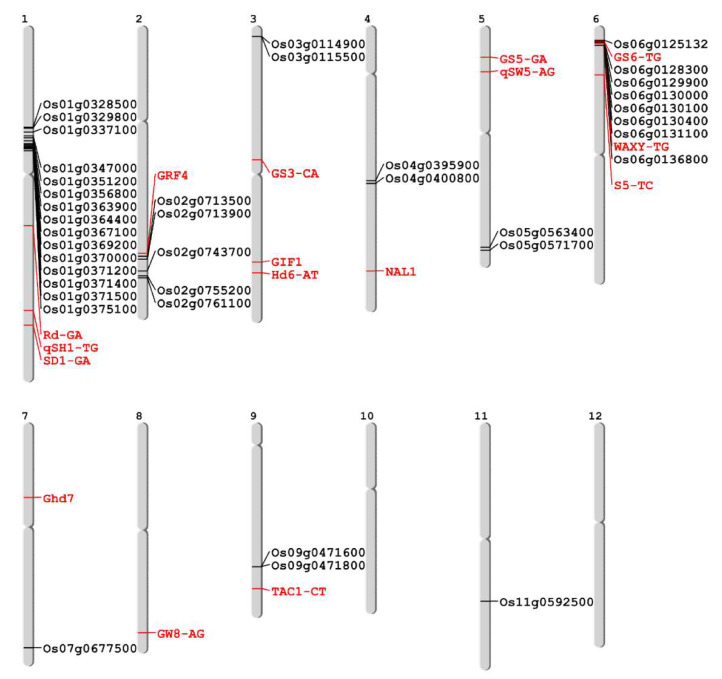
Genomic location of 54 polymorphic SNP markers used in this study. The markers represented by black and red indicate those newly developed on common *japonica* regions and those previously developed for agronomic trait related genes, respectively.

**Figure 3 genes-11-00562-f003:**
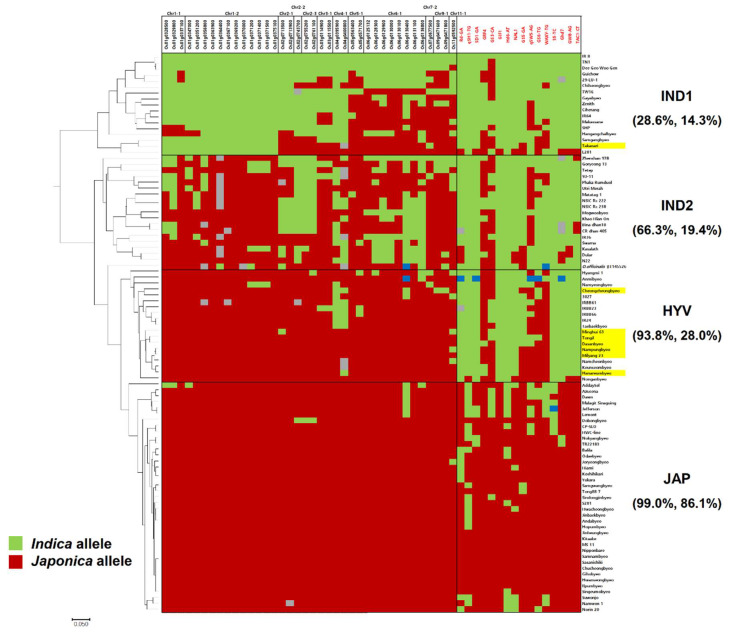
Phylogenetic tree of 94 germplasms based on 54 SNP markers and a genotype heat map. A 38 SNP marker set (left) and 16 SNP marker set (right) were developed on common *japonica* regions and agronomic trait related genes, respectively. The color of the marker ID is the same as in Figure 2. The varieties highlighted with yellow are eight resequenced HYVs. Homozygous alleles, which are identical to Nipponbare, were represented as red, and this is different from Nipponbare, represented as green and heterozygous alleles as blue. Grey indicates a missing genotype. The percentage values in parenthesis under each subgroup represent the percentage for homozygous *indica* allele in the 38 SNP marker set and 16 SNP marker set.

**Table 1 genes-11-00562-t001:** Basic sequencing statistics of the varieties used in this study.

Type	Variety Name	Yield (bp)	Read	N (%)	GC (%)	Q30 (%)	Depth (X)	Sequencing Platform
HYV	Cheongcheongbyeo	9,991,040,272	66,464,246	0.07	43.83	83.37	21.76	Illumina NextSeq 500
HYV	Dasanbyeo	5,677,243,407	93,838,734	1.05	39.75	69.57	10.32	Illumina HiSeq 1000
HYV	Hanareumbyeo	8,701,463,207	57,884,660	0.07	43.74	81.74	18.58	Illumina NextSeq 500
HYV	Milyang 23	8,909,120,495	147,258,190	0.31	41.08	76.44	17.79	Illumina HiSeq 1000
HYV	Minghui 63	10,346,919,646	68,794,138	0.07	43.93	83.18	22.48	Illumina NextSeq 500
HYV	Nampungbyeo	10,366,586,498	68,923,446	0.07	43.88	83.52	22.62	Illumina NextSeq 500
HYV	Takanari	9,372,371,998	62,353,794	0.08	43.38	83.10	20.35	Illumina NextSeq 500
HYV	Tongil	13,362,670,165	264,607,330	0.16	42.57	70.41	24.58	Illumina HiSeq 1000
*Japonica*	Nipponbare	22,212,867,380	439,858,760	0.19	42.10	89.12	51.72	Illumina HiSeq 1000
*Japonica*	Yukara	9,155,887,048	151,336,976	0.28	41.13	77.37	18.51	Illumina HiSeq 1000
*Indica*	IR 8	8,287,794,812	136,988,344	0.40	41.14	78.94	17.09	Illumina HiSeq 1000
*Indica*	TN 1	8,337,247,875	137,805,750	0.31	41.51	77.25	16.83	Illumina HiSeq 1000

Abbreviations are as follow: HYV—high-yielding variety, N (%)—percentage of skipped base, GC (%)—percentage of GC content, Q30 (%)—percentage of bases showing Phred quality score (Q) ≥ 30.

**Table 2 genes-11-00562-t002:** Number and location of the SNPs in the varieties against Nipponbare pseudomolecule.

Varieties	Non-Synonymous	Synonymous	Intron	5′ UTR	3′ UTR	Intergenic	Total
Cheongcheongbyeo	19,193	17,439	21,671	7076	29,403	1,019,466	1,114,248
Dasanbyeo	18,534	16,944	21,247	7006	28,598	975,433	1,067,762
Hanareumbyeo	18,059	16,492	20,657	6825	27,766	937,990	1,027,789
Milyang 23	17,836	16,414	20,110	6628	27,271	933,778	1,022,037
Minghui 63	20,117	18,601	22,351	7350	30,731	1,042,153	1,141,303
Nampungbyeo	19,592	17,960	21,588	7195	29,286	1,005,553	1,101,174
Takanari	18,631	16,936	20,881	6887	28,157	966,035	1,057,527
Tongil	18,427	16,837	20,816	6750	27,021	952,421	1,042,272
IR 8	20,578	18,834	22,847	7494	31,152	1,061,383	1,162,288
TN1	20,171	18,179	22,095	7614	30,286	1,041,363	1,139,708

**Table 3 genes-11-00562-t003:** *Japonica*-type SNP frequency (%) of Tongil and the other HYVs at common *japonica*-type regions among the eight HYVs.

Region	Narrowed Range (Mb)	Size (Mb)	Tongil	Cheongcheongbyeo	Dasanbyeo	Hanareumbyeo	Milyang 23	Nampungbyeo	Minghui 63	Takanari	Type
Chr1-1	12.3 ~ 13.6	1.3	23.3	23.3	23.3	23.3	23.3	23.3	23.3	0.0	All (ex. Takanari)
Chr1-2	13.7 ~ 15.7	2	31.4	31.4	31.4	31.4	31.4	31.5	31.5	0.0	All (ex. Takanari)
Chr2-1	29.3 ~ 29.7	0.4	64.1	14.1	14.1	14.1	14.1	14.1	14.4	14.1	All
Chr2-2	31.0 ~ 31.2	0.2	46.5	1.1	1.1	1.1	1.1	1.1	2.1	1.1	All
Chr2-3	31.4 ~ 32.2	0.8	77.4	17.3	17.6	17.6	17.6	17.6	21.6	17.6	All
Chr3-1	0.8 ~ 0.9	0.1	73.9	19.3	16.6	16.6	73.9	73.9	19.2	73.9	All
Chr4-1	19.3 ~ 19.9	0.6	75.9	0.0	8.2	8.2	77.3	8.2	7.0	8.2	All (ex. Cheongcheong)
Chr4-2	20.3 ~ 20.5	0.2	87.9	0.0	12.8	12.8	12.8	5.1	0.3	5.1	All (ex. Cheongcheong)
Chr5-1	28.0 ~ 29.0	1	76.9	82.1	23.9	23.9	23.9	23.9	13.7	23.9	All
Chr6-1	1.0 2.0	1	55.2	51.7	70.1	70.1	70.1	70.1	70.1	70.1	All
Chr7-1	28.5 ~ 28.6	0.1	32.0	6.5	5.0	6.5	4.8	6.5	6.5	0.0	All (ex. Takanari)
Chr7-2	28.6 ~ 28.8	0.2	93.5	9.7	7.7	7.6	33.2	9.7	7.6	0.0	All (ex. Takanari)
Chr9-1	17.9 ~ 18.4	0.5	13.9	0.7	13.9	13.9	13.9	13.9	13.9	13.9	All
Chr11-1	22.5 ~ 22.6	0.1	25.3	65.6	25.3	25.3	25.3	0.0	0.0	12.9	All (ex. Nampung/Minghui63)

Abbreviation is as follows: ex.—except.

**Table 4 genes-11-00562-t004:** Classification of the reported QTLs co-location with common *japonica*-type regions in eight HYVs.

Region	Eating Quality	Abiotic Tolerance	Biotic Resistance	Yield-Related	Root	Flowering	Other	Total
Chr1-1/1-2	0	4	0	2	0	0	1	7
Chr2-1/2-2/2-3	8	6	3	5	1	1	1	25
Chr3-1	1	8	0	0	2	1	0	12
Chr4-1/4-2	0	2	0	4	1	0	1	8
Chr5-1	3	2	0	6	2	0	0	13
Chr6-1	10	1	1	0	1	2	0	15
Chr7-1/7-2	0	1	0	3	0	1	0	5
Chr9-1	0	4	2	7	0	0	0	13
Chr11-1	1	1	0	1	0	0	0	3
Total	23	29	6	28	7	5	3	101

**Table 5 genes-11-00562-t005:** Selected 39 genes in the common *japonica*-type regions.

Region	Gene (RAP DB)	Alternative Name	Known Function	
			UniProt	RAP DB
Chr1-1	*Os01g0328500*			Bucentaur or craniofacial development family protein
	*Os01g0329800*	*IAI1*		YT521-B-like protein family protein
	*Os01g0337100*	*OsTPS1*		Similar to Sesquiterpene synthase
Chr1-2	*Os01g0347000*	*OsPHS1b, OsPP4, OsSTA14*		Similar to PROPYZAMIDE-HTPERSENSITIVE 1
	*Os01g0348900*	*salT(sal1), SalT1, sal1, Sal1, SALT, ML, SalT, OsSalT*	Salt stress-induced protein	SalT gene product
	*Os01g0351200*	*PARP2-A*	Poly [ADP-ribose] polymerase 2-A	Similar to Poly
	*Os01g0356800*	*OsEnS-6*		Domain of unknown function DUF3406, chloroplast translocase domain containing protein
	*Os01g0363900*	*OsWAK5*		Similar to HASTY
	*Os01g0364400*	*OsRLCK35*		Protein kinase, catalytic domain domain containing protein
	*Os01g0367100*	*PHD1*		Chloroplast-localized UDP-glucose epimerase (UGE), Galactolipid biosynthesis for chloroplast membranes, Photosynthetic capability and carbon assimilate homeostasis (Os01t0367100-01); NAD(P)-binding domain containing protein
	*Os01g0369200*	*CUL1*	Cullin-like protein	Similar to Cullin-1
	*Os01g0370000*	*OsOPR9, OsOPR01-2*	Putative 12-oxophytodienoate reductase 9	NADH:flavin oxidoreductase/NADH oxidase, N-terminal domain containing protein
	*Os01g0371200*	*OsGSTF1, RGSI*	Probable glutathione S-transferase GSTF1	Similar to Glutathione-S-transferase 19E50
	*Os01g0371400*	*OsGSTF9*		Similar to Glutathione s-transferase gstf2
	*Os01g0371500*	*OsGSTF10*		Similar to Glutathione-S-transferase 19E50
	*Os01g0375100*	*OsDjC6*	DNAJ heat shock N-terminal domain-containing protein-like	Similar to DnAJ-like protein slr0093
Chr2-1	*Os02g0713500*	*OsFbox108, Os_F0236*		F-box domain, cyclin-like domain containing protein.
	*Os02g0713900*	*HMGR I, Hmg1, HMGR1, HMGR 1, OsHMGR1*	3-hydroxy-3-methylglutaryl-coenzyme A reductase 1	Similar to 3-hydroxy-3-methylglutaryl-coenzyme A reductase 1
Chr2-2	*Os02g0743700*			Similar to RING-H2 finger protein ATL1Q
Chr2-3	*Os02g0755200*	*OsHDMA702, HDMA702*	Lysine-specific histone demethylase 1 homolog 1	Similar to amine oxidase family protein
	*Os02g0761100*	*OsCYP40b, OsCYP-8, OsCYP40-2*		Similar to Cyclophilin-40 (Expressed protein)
Chr3-1	*Os03g0114900*		Mitochondrial import inner membrane translocase subunit Tim17 family protein, expressed	Similar to putaive mitochondrial inner membrane protein
	*Os03g0115500*			Similar to pyridoxine 5’-phosphate oxidase-related
Chr4-1	*Os04g0395900*			Polynucleotide adenylyltransferase region domain containing protein
	*Os04g0400800*			Heavy metal transport/detoxification protein domain containing protein
Chr5-1	*Os05g0563400*	*OsARF15, ETT1, OsETT1, ARF3b*	Auxin response factor 15	Similar to Auxin response factor 5
	*Os05g0571700*	*OsFbox282, Os_F0643*		Cyclin-like F-box domain containing protein
Chr6-1	*Os06g0125132*	*SDH8B*	Succinate dehydrogenase subunit 8B, mitochondrial	Conserved hypothetical protein
	*Os06g0128300*	*OsABCB23, OsISC32*		Similar to Mitochondrial half-ABC transporter, Similar to STA1 (STARIK 1); ATPase, coupled to transmembrane movement of substances
	*Os06g0129900*	*Cytochrome P450*	Cytochrome P450	Similar to Cytochrome P450 CYPD
	*Os06g0130000*	*LMR*		AAA-type ATPase, Defense response, Similar to Tobacco mosaic virus helicase domain-binding protein (Fragment)
	*Os06g0130100*	*OsSIK1, OsER2, ER2*		Receptor-like kinase (RLK), Drought and salt stress tolerance, Oryza sativa stress-induced protein kinase gene 1
	*Os06g0130400*	*OsACS6*		ACC synthase, Protein homologous to aminotransferase, Ethylene biosynthesis, Control of starch grain size in rice endosperm
	*Os06g0131100*	*OsWD40-124*		Similar to guanine nucleotide-binding protein beta subunit-like protein 1, WD40/YVTN repeat-like domain containing protein
	*Os06g0136800*	*OsClp9, CLP9*	ATP-dependent Clp protease proteolytic subunit	Peptidase S14, ClpP family protein
Chr7-2	*Os07g0677500*	*POX3006, prx114*	Peroxidase	Similar to Peroxidase precursor (EC 1.11.1.7)
Chr9-1	*Os09g0471600*	*OsWAK84*		EGF-like calcium-binding domain containing protein
	*Os09g0471800*	*OsWAK85, YK10*		Similar to WAK80 - OsWAK receptor-like protein kinase
Chr11-1	*Os11g0592500*		NB-ARC domain containing protein, expressed	Similar to NB-ARC domain containing protein, expressed

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
