# Peer review of "Characterization of the Common Japonica-Originated Genomic Regions in the High-Yielding Varieties Developed from Inter-Subspecific Crosses in Temperate Rice (Oryza sativa L.)"

_genes, 2020, doi:10.3390/genes11050562_

Round 1

Reviewer 1 Report

In this manuscript, the authors identified a total of 14 common regions of japonica-originated alleles by a comparative study of the genomic regions in the eight HYVs, and, HYVs shared the japonica-originated regions on the nine chromosomes. In addition, the authors classified 94 germplasms into four groups with the 54 SNP markers. These results will contribute to the further development of more HYVs through inter-subspecific rice breeding.

However, some of the data feels insufficient to draw conclusions, and, the data in the table and the numbers in the text have many errors and doubts, so I should correct them.

*The page number in the upper right is not assigned correctly. The page numbers pointed out in the comments are renumbered from the beginning.

Major Comments:

  1. I could not understand the conclusions drawn by the Fig. 3. The authors describe that the markers can differentiate HYV from indica and japonica varieties. However, the HYV group includes several indica varieties, there is no explanation about those varieties. The 39 SNP markers used for analysis were selected by comparison of the eight HYVs, so I think it is natural to group them into HYV-group. Without information on other varieties classified as HYV-group, I don't think this analysis makes much sense.

  1. Table 4, I tried searching for some regions in the Q-TARO database, but the results did not match. In addition, the trait categories used in the Table 4 do not match in the Q-TARO database categorization. For example, ‘Yield related’ does not exist in the Q-TARO database, the authors should explain categorization.

  1. P. 8 L. 220, ‘’12 agronomic traits related SNP markers in indica-japonica SNP set 2’’. In the cited paper [28], ‘’14 functional or linked SNPs in genes previously cloned and associated with agronomic traits.’’ If there is a reason to reduce the number of markers, it should be described.

Minor Comments:

  1. In this manuscript, the classification of HYV is an important point, I think it is better to specify the definition of HYV.

  1. P.1 L. 23, it is described as classified by 39 SNP markers, but the actual classification is done with 38 markers, except one monomorphic SNP marker. I think it is appropriate to describe it as 38 markers.

  1. P. 5 L. 181, ‘’ at least six HYVs’’. Except for Takanari, Cheongcheong, and Minghui 63, I think that the number of HYVs is five. Please confirm.

  1. The legend of Figure 1, the vertical axis is not explained.

  1. In this manuscript, ‘Tongil-like’ appears, but it is described as ‘Tongil-type’ in the cited paper [2]. Are you changing it intentionally?

  1. It would be interesting to discuss the genomic regions that are not in Tongil but common to other HYVs. Please consider it.

  1. P.12 L. 270, is ‘’gens’’ a typo? Please confirm.

Author Response

Reviewer 1

Comments and Suggestions for Authors

In this manuscript, the authors identified a total of 14 common regions of japonica-originated alleles by a comparative study of the genomic regions in the eight HYVs, and, HYVs shared the japonica-originated regions on the nine chromosomes. In addition, the authors classified 94 germplasms into four groups with the 54 SNP markers. These results will contribute to the further development of more HYVs through inter-subspecific rice breeding. However, some of the data feels insufficient to draw conclusions, and, the data in the table and the numbers in the text have many errors and doubts, so I should correct them.

-> We appreciate the critical review and helpful suggestions, and the manuscript was revised according to the comments.

*The page number in the upper right is not assigned correctly. The page numbers pointed out in the comments are renumbered from the beginning.

-> The page numbers were corrected.

Major Comments:

I could not understand the conclusions drawn by the Fig. 3. The authors describe that the markers can differentiate HYV from indica and japonica varieties. However, the HYV group includes several indica varieties, there is no explanation about those varieties. The 39 SNP markers used for analysis were selected by comparison of the eight HYVs, so I think it is natural to group them into HYV-group. Without information on other varieties classified as HYV-group, I don't think this analysis makes much sense.

-> The HYV group in Figure 3 contains not only eight HYVs used for genome sequencing, but also seven other Tongil-type Korean varieties and five indica accessions (Table S1). Those varieties, which were clustered in the HYV group, were developed by inter-subspecific crosses or by inter-varietal crosses, including the parents of Tongil-type varieties. For example, Keunseombyeo was derived from a cross between Dasanbyeo and Namyeongbyeo. Additionally, Taebaekbyeo was used for the development of Hanareumbyeo and Dasanbyeo (Figure S1). On the other hand, among five indica varieties, four indica (IR24, IRBB23, IRBB61, and IRBB66) were developed by International Rice Research Institute (IRRI). IRBB23, IRBB61, and IRBB66 are near-isogenic line series of IR24 for the improvement of resistance to bacterial blight. IR24 is a good eating quality and high-yielding variety, developed from inter-subspecific crosses using one tropical japonica (CP-SLO) and two indica (SIGADIS and IR8). Furthermore, IR24 was included in the breeding programs of six HYVs, which were used for resequencing in this study. (Figure S1, C-H). This implies that the specific genomic regions were conserved in HYV group by the selections during the conventional breeding programs for HYV development. We revised the result section for Figure 3 according to the explanation above.

Table 4, I tried searching for some regions in the Q-TARO database, but the results did not match. In addition, the trait categories used in the Table 4 do not match in the Q-TARO database categorization. For example, ‘Yield related’ does not exist in the Q-TARO database, the authors should explain categorization.

-> We extracted the QTL information from the Q-TARO database at August 25, 2017. The physical position in Q-TARO is based on IRGSP build 4 of the Nipponbare genome, while IRGSP 1.0 of the Nipponbare genome was used in this study. Thus, the physical position of the start and end of each QTLs in Q-TARO were converted into the physical positions of IRGSP 1.0. Then, the QTLs overlapping on common japonica-originated regions of eight HYVs were selected. After checking if all the selected QTLs were unique or redundant, the redundant QTLs were discarded and only the unique QTLs were left remained. After the filtering step for the QTLs, they were classified into seven categories as shown in Table 4. The category classification was conducted by checking the character and trait name manually. For instance, the ‘Yield-relate’ category contained various characters such as source activity, sink related morphological traits, and sterility. For this reason, the ‘yield-related’ trait category of Table 4 was not found in the Q-TARO database. We revised the manuscript according to the comment mentioned above (Material and methods, 2.6 and Results page 18.)        

  1. 8 L. 220, ‘’12 agronomic traits related SNP markers in indica-japonica SNP set 2’’. In the cited paper [28], ‘’14 functional or linked SNPs in genes previously cloned and associated with agronomic traits.’’ If there is a reason to reduce the number of markers, it should be described.

-> The two SNP markers were SaF-CT and SLG7-GC. Those markers showed low base call quality, and excluded from further analysis. The result 3.4 part of the manuscript was revised.

Minor Comments:

In this manuscript, the classification of HYV is an important point, I think it is better to specify the definition of HYV.

-> The definition of HYV is not easily defined in terms of a definite amount of production. They perform high yield in specific environmental conditions. For this reason, Tongil was firstly considered in our previous paper. It performed around 30% higher yield than the common varieties in Korea (Line 42-43). So, we would like to reserve the specified term for ‘HYV’ after collecting more concrete information. With the same reason, we simply mentioned that HYVs developed from indica-japonica crosses in this study (line 40-41), which means that HYVs do not need to be developed only by inter-subspecific crosses. We put ‘temperate’ in the title.  

P.1 L. 23, it is described as classified by 39 SNP markers, but the actual classification is done with 38 markers, except one monomorphic SNP marker. I think it is appropriate to describe it as 38 markers.

-> It was revised to “38 single nucleotide polymorphism (SNP) markers from 38 genes”.

  1. 5 L. 181, ‘’ at least six HYVs’’. Except for Takanari, Cheongcheong, and Minghui 63, I think that the number of HYVs is five. Please confirm.

-> The sentence means we had certain region shared in at least six HYVs on the japonica-originated segment on Tongil.

The legend of Figure 1, the vertical axis is not explained.

-> If the vertical axis means black dotted line, it represents ‘the position and range for the co-location of the japonica-type blocks’. We revised the last sentence in the legend of Figure 1.

In this manuscript, ‘Tongil-like’ appears, but it is described as ‘Tongil-type’ in the cited paper [2]. Are you changing it intentionally?

-> ‘Tongil-like’ were revised to ‘Tongil-type’ in the cases used for varieties developed by inter-subspecific crosses in Korea like reference paper. However, ‘Tongil-like’ was used for meaning of ‘similar to Tongil’.

It would be interesting to discuss the genomic regions that are not in Tongil but common to other HYVs. Please consider it.

-> Thank you for your comments. We were supposed to do it, however, the analysis about the regions were for a following study, as it will be better to report together with the functional roles of japonica-originated regions in many indica-japonica HYVs. In this paper, we are focusing on the regions originated from the well-known and historically validated variety of Tongil first.

P.12 L. 270, is ‘’gens’’ a typo? Please confirm.

-> It was corrected to “genes”.

Reviewer 2 Report

  1. In abstract (L20-21), there is no evidence to support in the sentence. The reads had better be aligned to the indica reference (e.g. R498) at first. Then non-indica reads (including unmapped reads) would be aligned to the IRGSP-1.0. The inter-subspecific cultivars are closely related, and some neutral regions as if desirable regions.
  2. In introduction (L49-54), “Minghui 63” is an indica rice (Zhang et al., PNAS 113: E5163).
  3. In results (Table 1), “Sequencing method” may be replaced with “Sequencing platform”.
  4. In results (170-171), the sentences had better place in discussion.
  5. In results (Table 2, Figure S1A, S1H), “Taichung Native 1” may be replaced with “TN1”.
  6. In discussion (L296), “scientists” may be replaced with “breeders”.
  7. In conclusion, it gives off an erratic impression.

Author Response

Reviewer 2

Comments and Suggestions for Authors

In abstract (L20-21), there is no evidence to support in the sentence. The reads had better be aligned to the indica reference (e.g. R498) at first. Then non-indica reads (including unmapped reads) would be aligned to the IRGSP-1.0. The inter-subspecific cultivars are closely related, and some neutral regions as if desirable regions.

-> Thank you for your comment. It will be great to map the whole sequence to indica as well. However, in this study, we focused to dissect the japonica-originated regions in Tongil by comparing common Tongil-like japonica regions in HYVs. For this reason, the NGS mapping was conducted on Nipponbare (japonica reference) genome using SNP allele calling method described in Materials and Methods. Actually, we have conducted the mapping with indica (Zhenshan 97), however, with a high level of polymorphism in all the chromosomal regions including our target regions. It was more difficult to define the borders of the introgressions. The variations focused in the paper is majorly introgression of japonica regions to indica, so mapping to the standard japonica reference could be fine in this study.

In introduction (L49-54), “Minghui 63” is an indica rice (Zhang et al., PNAS 113: E5163).

-> “Minghui 63 was classified an indica II subpopulation together with Milyang 23 and some other indica-japonica HYVs by genome sequence analysis [8].” was added on last part of that paragraph.

In results (Table 1), “Sequencing method” may be replaced with “Sequencing platform”.

-> It was revised.

In results (170-171), the sentences had better place in discussion.

-> This is a kind of short implication. We do not have any more comments on this. In general, we understand that a short comment can be allowed in the result.

In results (Table 2, Figure S1A, S1H), “Taichung Native 1” may be replaced with “TN1”.

-> Those were replaced to “TN1”

In discussion (L296), “scientists” may be replaced with “breeders”.

-> It was revised.

In conclusion, it gives off an erratic impression.

-> We are sorry that we could not get your point. We are ready to listen to your critical idea if you have. We have a thoughtful mind in our hands.

Reviewer 3 Report

The authors examined the common genomic regions among HYVs, which were derived from japonica genome. This study includes novel findings and important genomic implications for further development of high yielding rice varieties. The experiments were appropriately designed. I had no specific queries except of some small questions.

Table 1 Is the number of read represented by “bp” unit?

L170 “This implies that most of HYVs, except for Minghui 63, possess some genomic segments inherited from japonica.”

I didn’t totally understand why the result of table2 led to the conclusion of most of HYVs possessing some genomic segments inherited from japonica, and why Minghui 63 should be excluded. Please add logical explanation for this conclusion.

L248 “By the way, high proportion of japonica alleles on Chr1-1, Chr1-2, and Chr3-1 was found in IND2, which consist of many varieties known as aus variety group. “

This sentence seems unclear. Make modification with some numeric values supporting results.

L278 PHD shoule be PHD1

Author Response

Reviewer 3

Comments and Suggestions for Authors

The authors examined the common genomic regions among HYVs, which were derived from japonica genome. This study includes novel findings and important genomic implications for further development of high yielding rice varieties. The experiments were appropriately designed. I had no specific queries except of some small questions.

-> We appreciate your kind review and the manuscript was revised according to the comments.

Table 1 Is the number of read represented by “bp” unit?

-> The number of read means number of short sequence read generated from NGS. The length of read was different depend on sequencing platform. We removed ‘bp’ from the table. It was our mistake. Sorry about that.

L170 “This implies that most of HYVs, except for Minghui 63, possess some genomic segments inherited from japonica.”

I didn’t totally understand why the result of table2 led to the conclusion of most of HYVs possessing some genomic segments inherited from japonica, and why Minghui 63 should be excluded. Please add logical explanation for this conclusion.

-> Thank you for your comment. We agree and the sentence was removed.

L248 “By the way, high proportion of japonica alleles on Chr1-1, Chr1-2, and Chr3-1 was found in IND2, which consist of many varieties known as aus variety group. “

This sentence seems unclear. Make modification with some numeric values supporting results.

-> We rechecked varieties in the IND2 subgroup whether each was aus or not. There were only three exact aus varieties: Kasalath, Dular, and N22. Furthermore, one wild-rice accession and one weedy rice accession were clustered in IND2. We have revised the sentence accordingly.

L278 PHD shoule be PHD1

-> It was corrected.

Round 2

Reviewer 2 Report

> Actually, we have conducted the mapping with indica (Zhenshan 97), however, with a high level of polymorphism in all the chromosomal regions including our target regions. It was more difficult to define the borders of the introgressions.

That is right. How do you know you are right?